# Easy Differentially Private Linear Regression

**Kareem Amin**  **Matthew Joseph**  **Mónica Ribero**  **Sergei Vassilvitskii**[*]

## Abstract

Linear regression is a fundamental tool for statistical analysis. This has motivated the development of linear regression methods that also satisfy differential privacy and thus guarantee that the learned model reveals little about any one data point used to construct it. However, existing differentially private solutions assume that the end user can easily specify good data bounds and hyperparameters. Both present significant practical obstacles. In this paper, we study an algorithm which uses the exponential mechanism to select a model with high Tukey depth from a collection of non-private regression models. Given $n$ samples of $d$-dimensional data used to train $m$ models, we construct an efficient analogue using an approximate Tukey depth that runs in time $O(d^2 n + dm \log(m))$. We find that this algorithm obtains strong empirical performance in the data-rich setting with no data bounds or hyperparameter selection required.

## 1 Introduction

Existing methods for differentially private linear regression include objective perturbation (Kifer et al., 2012), ordinary least squares (OLS) using noisy sufficient statistics (Dwork et al., 2014; Wang, 2018; Sheffet, 2019), and DP-SGD (Abadi et al., 2016). Carefully applied, these methods can obtain high utility in certain settings. However, each method also has its drawbacks. Objective perturbation and sufficient statistics require the user to provide bounds on the feature and label norms, and DP-SGD requires extensive hyperparameter tuning (of clipping norm, learning rate, batch size, and so on).

In practice, users of differentially private algorithms struggle to provide instance-specific inputs like feature and label norms without looking at the private data (Sarathy et al., 2022). Unfortunately, looking at the private data also nullifies the desired differential privacy guarantee. Similarly, while recent work has advanced the state of the art of private hyperparameter tuning (Liu & Talwar, 2019; Papernot & Steinke, 2022), non-private hyperparameter tuning remains the most common and highest utility approach in practice. Even ignoring its (typically elided) privacy cost, this tuning adds significant time and implementation overhead. Both considerations present obstacles to differentially private linear regression in practice.

With these challenges in mind, the goal of this work is to provide an easy differentially private linear regression algorithm that works quickly and with no user input beyond the data itself. Here, "ease" refers to the experience of end users. The algorithm we propose requires care to construct and implement, but it only requires an end user to specify their dataset and desired level of privacy. We also emphasize that ease of use, while nice to have, is not itself the primary goal. Ease of use affects both privacy and utility, as an algorithm that is difficult to use will sacrifice one or both when data bounds and hyperparameters are imperfectly set.

### 1.1 Contributions

Our algorithm generalizes previous work by Alabi et al. (2022), which proposes a differentially private variant of the Theil-Sen estimator for one-dimensional linear regression (Theil, 1992). The core idea is to partition the data into $m$ subsets, non-privately estimate a regression model on each, and then apply the exponential mechanism with some notion of depth to privately estimate a high-depth model from a restricted domain that the end user specifies. In the simple one-dimensional case (Alabi et al.,

---

[*]{kamin, mtjoseph, mribero, sergeiv}@google.com. Part of this work done while Mónica was at UT Austin.

2022) each model is a slope, the natural notion of high depth is the median, and the user provides an interval for candidate slopes.

We generalize this in two ways to obtain our algorithm, TukeyEM. The first step is to replace the median with a multidimensional analogue based on Tukey depth. Second, we adapt a technique based on propose-test-release (PTR), originally introduced by Brown et al. (2021) for private estimation of unbounded Gaussians, to construct an algorithm which does not require bounds on the domain for the overall exponential mechanism. We find that a version of TukeyEM using an approximate and efficiently computable notion of Tukey depth achieves empirical performance competitive with (and often exceeding) that of *non-privately tuned* baseline private linear regression algorithms, across several synthetic and real datasets. We highlight that the approximation only affects utility and efficiency; TukeyEM is still differentially private. Given an instance where TukeyEM constructs $m$ models from $n$ samples of $d$-dimensional data, the main guarantee for our algorithm is the following:

**Theorem 1.1.** TukeyEM *is $(\varepsilon, \delta)$-DP and takes time $O(d^2 n + dm \log(m))$.*

Two caveats apply. First, our use of PTR comes at the cost of an approximate $(\varepsilon, \delta)$-DP guarantee as well as a failure probability: depending on the dataset, it is possible that the PTR step fails, and no regression model is output. Second, the algorithm technically has one hyperparameter, the number $m$ of models trained. Our mitigation of both issues is empirical. Across several datasets, we observe that a simple heuristic about the relationship between the number of samples $n$ and the number of features $d$, derived from synthetic experiments, typically suffices to ensure that the PTR step passes and specifies a high-utility choice of $m$. For the bulk of our experiments, the required relationship is on the order of $n \gtrsim 1000 \cdot d$. We emphasize that this heuristic is based only on the data dimensions $n$ and $d$ and does not require further knowledge of the data itself.

## 1.2 RELATED WORK

Linear regression is a specific instance of the more general problem of convex optimization. Ignoring dependence on the parameter and input space diameter for brevity, DP-SGD (Bassily et al., 2014) and objective perturbation (Kifer et al., 2012) obtain the optimal $O(\sqrt{d}/\varepsilon)$ error for empirical risk minimization. AdaOPS and AdaSSP also match this bound (Wang, 2018). Similar results are known for population loss (Bassily et al., 2019), and still stronger results using additional statistical assumptions on the data (Cai et al., 2020; Varshney et al., 2022). Recent work provides theoretical guarantees with no boundedness assumptions on the features or labels (Milionis et al., 2022) but requires bounds on the data's covariance matrix to use an efficient subroutine for private Gaussian estimation and does not include an empirical evaluation. The main difference between these works and ours is empirical utility without data bounds and hyperparameter tuning.

Another relevant work is that of Liu et al. (2022), which also composes a PTR step adapted from Brown et al. (2021) with a call to a restricted exponential mechanism. The main drawback of this work is that, as with the previous work (Brown et al., 2021), neither the PTR step nor the restricted exponential mechanism step is efficient. This applies to other works that have applied Tukey depth to private estimation as well (Beimel et al., 2019; Kaplan et al., 2020; Liu et al., 2021; Ramsay & Chenouri, 2021). The main difference between these works and ours is that our approach produces an efficient, implemented mechanism.

Finally, concurrent independent work by Cumings-Menon (2022) also studies the usage of Tukey depth, as well as the separate notion of regression depth, to privately select from a collection of non-private regression models. A few differences exist between their work and ours. First, they rely on additive noise scaled to smooth sensitivity to construct a private estimate of a high-depth point. Second, their methods are not computationally efficient beyond small $d$, and are only evaluated for $d \leq 2$. Third, their methods require the end user to specify bounds on the parameter space.

## 2 PRELIMINARIES

We start with the definition of differential privacy, using the "add-remove" variant.

**Definition 2.1** (Dwork et al. (2006))**.** *Databases $D, D'$ from data domain $\mathcal{X}$ are* neighbors*, denoted $D \sim D'$, if they differ in the presence or absence of a single record. A randomized mechanism*

$\mathcal{M} : \mathcal{X} \to \mathcal{Y}$ *is* $(\varepsilon, \delta)$-*differentially private (DP) if for all* $D \sim D' \in \mathcal{X}$ *and any* $S \subseteq \mathcal{Y}$

$$\mathbb{P}_{\mathcal{M}} \left[ \mathcal{M}(D) \in S \right] \leq e^{\varepsilon} \mathbb{P}_{\mathcal{M}} \left[ \mathcal{M}(D') \in S \right] + \delta.$$

When $\delta = 0$, $\mathcal{M}$ is $\varepsilon$-DP. One general $\varepsilon$-DP algorithm is the exponential mechanism.

**Definition 2.2** (McSherry & Talwar (2007))**.** *Given database* $D$ *and utility function* $u : \mathcal{X} \times \mathcal{Y} \to \mathbb{R}$ *mapping* (*database, output*) *pairs to scores with sensitivity*

$$\Delta_u = \max_{D \sim D', y \in \mathcal{Y}} |u(D, y) - u(D', y)|,$$

*the* exponential mechanism *selects item* $y \in \mathcal{Y}$ *with probability proportional to* $\exp \left( \frac{\epsilon u(D,y)}{2\Delta_u} \right)$. *We say the utility function* $u$ *is* monotonic *if, for* $D_1 \subset D_2$, *for any* $y$, $u(D_1, y) \leq u(D_2, y)$. *Given monotonic* $u$, *the 2 inside the exponent denominator can be dropped.*

**Lemma 2.3** (McSherry & Talwar (2007))**.** *The exponential mechanism is* $\epsilon$-*DP.*

Finally, we define Tukey depth.

**Definition 2.4** (Tukey (1975))**.** *A* halfspace $h_v$ *is defined by a vector* $v \in \mathbb{R}^d$, $h_v = \{y \in \mathbb{R}^d \mid \langle v, y \rangle \geq 0\}$. *Let* $D \subset \mathbb{R}^d$ *be a collection of* $n$ *points. The* Tukey depth $T_D(y)$ *of a point* $y \in \mathbb{R}^d$ *with respect to* $D$ *is the minimum number of points in* $D$ *in any halfspace containing* $y$,

$$T_D(y) = \min_{h_v \mid y \in h_v} \sum_{x \in D} \mathbb{1}_{x \in h_v}.$$

Note that for a collection of $n$ points, the maximum possible Tukey depth is $n/2$. We will prove a theoretical utility result for a version of our algorithm that uses exact Tukey depth. However, Tukey depth is NP-hard to compute for arbitrary $d$ (Johnson & Preparata, 1978), so our experiments instead use a notion of approximate Tukey depth that can be computed efficiently. The approximate notion of Tukey depth only takes a minimum over the $2d$ halfspaces corresponding to the canonical basis.

**Definition 2.5.** *Let* $E = \{e_1, ..., e_d\}$ *be the canonical basis for* $\mathbb{R}^d$ *and let* $D \subset \mathbb{R}^d$. *The* approximate Tukey depth *of a point* $y \in \mathbb{R}^d$ *with respect to* $D$, *denoted* $\tilde{T}_D(y)$, *is the minimum number of points in* $D$ *in any of the* $2d$ *halfspaces determined by* $E$ *containing* $y$,

$$\tilde{T}_D(y) = \min_{e_j \mid e_j \in \pm E, y \in h_{y_i \cdot e_j}} \sum_{x \in D} \mathbb{1}_{x \in h_{y_i \cdot e_j}}.$$

Stated more plainly, approximate Tukey depth only evaluates depth with respect to the $d$ axis-aligned directions. A simple illustration of the difference between exact and approximate Tukey depth appears in Figure 3 in the Appendix's Section 7.1. For both exact and approximate Tukey depth, when $D$ is clear from context, we omit it for neatness.

## 3 MAIN ALGORITHM

Our algorithm, TukeyEM, consists of four steps:

1. Randomly partition the dataset into $m$ subsets, non-privately compute the OLS estimator on each subset, and collect the $m$ estimators into set $\{\beta_i\}_{i=1}^m$.
2. Compute the volumes of regions of different approximate Tukey depths with respect to $\{\beta_i\}_{i=1}^m$.
3. Run a propose-test-release (PTR) algorithm using these volumes. If it passes, set $B$ to be the region of $\mathbb{R}^d$ with approximate Tukey depth at least $m/4$ in $\{\beta_i\}_{i=1}^m$ and proceed to the next step. If not, release $\perp$ (failure).
4. If the previous step succeeds, apply the exponential mechanism, using approximate Tukey depth as the utility function, to privately select a point from $B$.

A basic utility result for the version of TukeyEM using exact Tukey depth appears below. The result is a direct application of work from Brown et al. (2021), and the (short) proof appears in the Appendix's Section 7.2..

**Theorem 3.1** (Brown et al. (2021)). *Let $0 < \alpha, \gamma < 1$ and let $S = \{\beta_1, ..., \beta_m\}$ be an i.i.d. sample from the multivariate normal distribution $\mathcal{N}(\beta^*, \Sigma)$ with covariance $\Sigma \in \mathbb{R}^{d \times d}$ and mean $\mathbb{E}[\beta_i] = \beta^* \in \mathbb{R}^d$. Given $\hat{\beta} \in \mathbb{R}^d$ with Tukey depth at least $p$ with respect to $S$, there exists a constant $c > 0$ such that when $m \geq c\left(\frac{d + \log(1/\gamma)}{\alpha^2}\right)$ with probability $1 - \gamma$, $\|\hat{\beta} - \beta^*\|_\Sigma \leq \Phi^{-1}(1 - p/m + \alpha)$, where $\Phi$ denotes the CDF of of the standard univariate Gaussian.*

In practice, we observe that empirical distributions of models for real data often feature Gaussian-like concentration, fast tail decay, and symmetry. Plots of histograms for the the models learned by TukeyEM on experiment datasets appear in the Appendix's Section 7.7. Nonetheless, we emphasize that Theorem 3.1 is a statement of sufficiency, not necessity. TukeyEM does not require any distributional assumption to be private, nor does non-Gaussianity preclude accurate estimation.

The remaining subsections elaborate on the details of our version using approximate Tukey depth, culminating in the full pseudocode in Algorithm 2 and overall result, Theorem 1.1.

### 3.1 COMPUTING VOLUMES

We start by describing how to compute volumes corresponding to different Tukey depths. As shown in the next subsection, these volumes will be necessary for the PTR subroutine.

**Definition 3.2.** *Given database $D$, define $V_{i,D} = \mathsf{vol}(\{y \mid y \in \mathbb{R}^d \text{ and } \tilde{T}_D(y) \geq i\})$, the volume of the region of points in $\mathbb{R}^d$ with approximate Tukey depth at least $i$ in $D$. When $D$ is clear from context, we write $V_i$ for brevity.*

Since our notion of approximate Tukey depth uses the canonical basis (Definition 2.5), it follows that $V_1, V_2, \ldots, V_{m/2}$[1] form a sequence of nested (hyper)rectangles, as shown in Figure 3. With this observation, computing a given $V_i$ is simple. For each axis, project the non-private models $\{\beta_i\}_{i=1}^m$ onto the axis and compute the distance between the two points of exact Tukey depth $i$ (from the "left" and "right") in the one-dimensional sorted array. This yields one side length for the hyperrectangle. Repeating this $d$ times in total and taking the product then yields the total volume of the hyperrectangle, as formalized next. The simple proof appears in the Appendix's Section 7.2.

**Lemma 3.3.** *Lines 5 to 10 of Algorithm 2 compute $\{V_i\}_{i=1}^{m/2}$ in time $O(dm \log(m))$.*

### 3.2 APPLYING PROPOSE-TEST-RELEASE

The next step of TukeyEM employs PTR to restrict the output region eventually used by the exponential mechanism. We collect this process into a subroutine PTRCheck.

The overall strategy applies work done by Brown et al. (2021). Their algorithm privately checks if the given database has a large Hamming distance to any "unsafe" database and then, if this PTR check passes, runs an exponential mechanism restricted to a domain of high Tukey depth. Since a "safe" database is defined as one where the restricted exponential mechanism has a similar output distribution on any neighboring database, the overall algorithm is DP. As part of their utility analysis, they prove a lemma translating a volume condition on regions of different Tukey depths to a lower bound on the Hamming distance to an unsafe database (Lemma 3.8 (Brown et al., 2021)). This enables them to argue that the PTR check typically passes if it receives enough Gaussian data, and the utility guarantee follows. However, their algorithm requires computing both exact Tukey depths of the samples and the current database's exact Hamming distance to unsafety. The given runtimes for both computations are exponential in the dimension $d$ (see their Section C.2 (Brown et al., 2021)).

We rely on approximate Tukey depth (Definition 2.5) to resolve both issues. First, as the previous section demonstrated, computing the approximate Tukey depths of a collection of $m$ $d$-dimensional points only takes time $O(dm \log(m))$. Second, we adapt their lower bound to give a *1-sensitive* lower bound on the Hamming distance between the current database and any unsafe database. This yields an efficient replacement for the exact Hamming distance calculation used by Brown et al. (2021).

The overall structure of PTRCheck is therefore as follows: use the volume condition to compute a 1-sensitive lower bound on the given database's distance to unsafety; add noise to the lower bound

---

[1]We assume $m$ is even for simplicity. The algorithm and its guarantees are essentially the same when $m$ is odd, and our implementation handles both cases.

---

**Algorithm 1** PTRCheck

---

1: **Input:** Tukey depth region volumes $V$, privacy parameters $\varepsilon$ and $\delta$
2: Use Lemma 3.6 with $t = \frac{|V|}{2}$ and $\frac{\delta}{8e^\varepsilon}$ to compute lower bound $k$ for distance to unsafe database
3: **if** $k + \mathtt{Lap}\,(1/\varepsilon) \geq \frac{\log(1/2\delta)}{\varepsilon}$ **then**
4:     Return True
5: **else**
6:     Return False

---

and compare it to a threshold calibrated so that an unsafe dataset has probability $\leq \delta$ of passing; and if the check passes, run the exponential mechanism to pick a point of high approximate Tukey depth from the domain of points with moderately high approximate Tukey depth. Before proceeding to the details of the algorithm, we first define a few necessary terms.

**Definition 3.4** (Definition 2.1 Brown et al. (2021)). *Two distributions $\mathcal{P}, \mathcal{Q}$ over domain $\mathcal{W}$ are $(\varepsilon, \delta)$-indistinguishable, denoted $\mathcal{P} \approx_{\varepsilon,\delta} \mathcal{Q}$, if for any measurable subset $W \subset \mathcal{W}$,*

$$\mathbb{P}_{w\sim\mathcal{P}}\left[w \in W\right] \leq e^\varepsilon \mathbb{P}_{w\sim\mathcal{Q}}\left[w \in W\right] + \delta \text{ and } \mathbb{P}_{w\sim\mathcal{Q}}\left[w \in W\right] \leq e^\varepsilon \mathbb{P}_{w\sim\mathcal{P}}\left[w \in W\right] + \delta.$$

Note that $(\varepsilon, \delta)$-DP is equivalent to $(\varepsilon, \delta)$-indistinguishability between output distributions on arbitrary neighboring databases. Given database $D$, let $A$ denote the exponential mechanism with utility function $\tilde{T}_D$ (see Definition 2.5). Given nonnegative integer $t$, let $A_t$ denote the same mechanism that assigns score $-\infty$ to any point with score $< t$, i.e., only samples from points of score $\geq t$. We will say a database is "safe" if $A_t$ is indistinguishable between neighbors.

**Definition 3.5** (Definition 3.1 Brown et al. (2021)). *Database $D$ is $(\varepsilon, \delta, t)$-safe if for all neighboring $D' \sim D$, we have $A_t(D) \approx_{\varepsilon,\delta} A_t(D')$. Let $\mathtt{Safe}_{(\varepsilon,\delta,t)}$ be the set of safe databases, and let $\mathtt{Unsafe}_{(\varepsilon,\delta,t)}$ be its complement.*

We now state the main result of this section, Lemma 3.6. Briefly, it modifies Lemma 3.8 from Brown et al. (2021) to construct a 1-sensitive lower bound on distance to unsafety.

**Lemma 3.6.** *Define $M(D)$ to be a mechanism that receives as input database $D$ and computes the largest $k \in \{0, \ldots, t-1\}$ such that there exists $g > 0$ where, for volumes $V$ defined using a monotonic utility function,*

$$\frac{V_{t-k-1,D}}{V_{t+k+g+1,D}} \cdot e^{-\varepsilon g/2} \leq \delta$$

*or outputs $-1$ if the inequality does not hold for any such $k$. Then for arbitrary $D$*

1. *$M$ is 1-sensitive, and*

2. *for all $z \in \mathtt{Unsafe}_{(\varepsilon,4e^\varepsilon\delta,t)}$, $d_H(D, z) > M(D)$.*

The proof of Lemma 3.6 appears in the Appendix's Section 7.2. Our implementation of the algorithm described by Lemma 3.6 randomly perturbs the models with a small amount of noise to avoid having regions with 0 volume. We note that this does not affect the overall privacy guarantee.

PTRCheck therefore runs the mechanism defined by Lemma 3.6, add Laplace noise to the result, and proceeds to the restricted exponential mechanism if the noisy statistic crosses a threshold. Pseudocode appears in Algorithm 1, and we now state its guarantee as proved in the Appendix's Section 7.2.

**Lemma 3.7.** *Given the depth volumes $V$ computed in Lines 9 to 10 of Algorithm 2, PTRCheck($V, \varepsilon, \delta$) is $\varepsilon$-DP and takes time $O(m \log(m))$.*

## 3.3 SAMPLING

If PTRCheck passes, TukeyEM then calls the exponential mechanism restricted to points of approximate Tukey depth at least $t = m/4$, a subroutine denoted RestrictedTukeyEM (Line 12 in Algorithm 2). Note that the passage of PTR ensures that with probability at least $1 - \delta$, running RestrictedTukeyEM is $(\varepsilon, \delta)$-DP. We use a common two step process for sampling from an exponential mechanism over a continuous space: 1) sample a depth using the exponential mechanism, then 2) return a uniform sample from the region corresponding to the sampled depth.

### 3.3.1 Sampling a Depth

We first define a slight modification $W$ of the volumes $V$ introduced earlier.

**Definition 3.8.** *Given database $D$, define $W_{i,D} = \mathsf{vol}(\{y \mid y \in \mathbb{R}^d \text{ and } \tilde{T}_D(y) = i\})$, the volume of the region of points in $\mathbb{R}^d$ with approximate Tukey depth exactly $i$ in $D$.*

To execute the first step of sampling, for $i \in \{m/4, m/4+1, \ldots, m/2\}$, $W_{i,D} = V_{i,D} - V_{i+1,D}$, so we can compute $\{W_{i,D}\}_{i=m/4}^{m/2}$ from the $V$ computed earlier in time $O(m)$. The restricted exponential mechanism then selects approximate Tukey depth $i \in \{m/4, m/4+1, \ldots, m/2\}$ with probability

$$\mathbb{P}[i] \propto W_{i,D} \cdot \exp(\varepsilon \cdot i).$$

Note that this expression drops the 2 in the standard exponential mechanism because approximate Tukey depth is monotonic; see Appendix Section 7.3 for details. For numerical stability, racing sampling Medina & Gillenwater (2020) can sample from this distribution using logarithmic quantities.

### 3.3.2 Uniformly Sampling From a Region

Having sampled a depth $\hat{i}$, it remains to return a uniform random point of approximate Tukey depth $\hat{i}$. By construction, $W_{\hat{i},D}$ is the volume of the set of points $y = (y_1, ..., y_d)$ such that the depth along every dimension $j$ is at least $\hat{i}$, and the depth along at least one dimension $j'$ is exactly $\hat{i}$. The result is straightforward when $d = 1$: draw a uniform sample from the union of the two intervals of points of depth exactly $\hat{i}$ (depth from the "left" and "right").

For $d > 1$, the basic idea of the sampling process is to partition the overall volume into disjoint subsets, compute each subset volume, sample a subset according to its proportion in the whole volume, and then sample uniformly from that subset. Our partition will split the overall region of depth exactly $i$ according to the first dimension with dimension-specific depth exactly $i$. Since any point in the overall region has at least one such dimension, this produces a valid partition, and we will see that computing the volumes of these partitions is straightforward using the $S$ computed earlier. Finally, the last sampling step will be easy because the final subset will simply be a pair of (hyper)rectangles. Since space is constrained and the details are relatively straightforward from the sketch above, full specification and proofs for this process $\mathsf{SamplePointWithDepth}(S, i)$ appear in Section 7.4. For immediate purposes, it suffices to record the following guarantee:

**Lemma 3.9.** $\mathsf{SamplePointWithDepth}(S, i)$ *returns a uniform random sample from the region of points with approximate Tukey depth $i$ in $S$ in time $O(d)$.*

### 3.4 Overall Algorithm

We now have all of the necessary material for the main result, Theorem 1.1, restated below. The proof essentially collects the results so far into a concise summary.

**Theorem 3.10.** $\mathsf{TukeyEM}$, *given in Algorithm 2, is $(\varepsilon, \delta)$-DP and takes time $O\left(d^2 n + dm \log(m)\right)$.*

*Proof.* Line 11 of the TukeyEM pseudocode in Algorithm 2 calls the check with privacy parameters $\varepsilon/2$ and $\delta/[8e^\varepsilon]$. By the sensitivity guarantee of Lemma 3.6, the check itself is $\varepsilon/2$-DP. By the safety guarantee of Lemma 3.6 and our choice of threshold, if it passes, with probability at least $1 - \delta/2$, the given database lies in $\mathsf{Safe}_{(\varepsilon/2, \delta/2, t)}$. A passing check therefore ensures that the sampling step in Line 12 is $(\varepsilon/2, \delta)$-DP. By composition, the overall privacy guarantee is $(\varepsilon, \delta)$-DP. Turning to runtime, the $m$ OLS computations inside Line 3 each multiply $d \times \frac{n}{m}$ and $\frac{n}{m} \times d$ matrices, for $O(d^2 n)$ time overall. From Lemma 3.3, Lines 5 to 10 take time $O(dm \log(m))$. Lemma 3.7 gives the $O(m \log(m))$ time for Line 11, and Lemma 3.9 gives the $O(d)$ time for Line 12. $\qquad\square$

## 4 Experiments

### 4.1 Baselines

    1. NonDP computes the standard non-private OLS estimator $\beta^* = (X^T X)^{-1} X^T y$.

---

**Algorithm 2** TukeyEM

---

1: **Input:** Features matrix $X \in \mathbb{R}^{n \times d}$, label vector $y \in \mathbb{R}^n$, number of models $m$, privacy parameters $\varepsilon$ and $\delta$
2: Evenly and randomly partition $X$ and $y$ into subsets $\{(X_i, y_i)\}_{i=1}^m$
3: **for** $i = 1, \ldots, m$ **do**
4:     Compute OLS estimator $\beta_i \leftarrow (X_i^T X_i)^{-1} X_i^T y_i$
5: **for** dimension $j \in [d]$ **do**
6:     $\{\beta_{i,j}\}_{i=1}^m \leftarrow$ projection of $\{\beta_i\}_{i=1}^m$ onto dimension $j$
7:     $(S_{j,1}, \ldots, S_{j,m}) \leftarrow \{\beta_{i,j}\}_{i=1}^m$ sorted in nondecreasing order
8: Collect projected estimators into $S \in \mathbb{R}^{d \times m}$, where each row is nondecreasing
9: **for** $i \in [m/2]$ **do**
10:     Compute volume of region of depth $\geq i$, $V_i \leftarrow \prod_{j=1}^d (S_{j,m-(i-1)} - S_{j,i})$
11: **if** PTRCheck$(V, \varepsilon/2, \delta)$ **then**
12:     $\hat{\beta} \leftarrow$ RestrictedTukeyEM$(V, S, m/4, \varepsilon/2)$
13:     Return $\hat{\beta}$
14: **else**
15:     Return $\perp$

---

2. AdaSSP (Wang, 2018) computes a DP OLS estimator based on noisy versions of $X^T X$ and $X^T y$. This requires the end user to supply bounds on both $\|X\|_2$ and $\|y\|_2$. Our implementation uses these values non-privately for each dataset. The implementation is therefore not private and represents an artificially strong version of AdaSSP. As specified by Wang (2018), AdaSSP (privately) selects a ridge parameter and runs ridge regression.

3. DPSGD (Abadi et al., 2016) uses DP-SGD, as implemented in TensorFlow Privacy and Keras (Chollet et al., 2015), to optimize mean squared error using a single linear layer. The layer's weights are regression coefficients. A discussion of hyperparameter selection appears in Section 4.4. As we will see, appropriate choices of these hyperparameters is both dataset-specific and crucial to DPSGD's performance. Since we allow DPSGD to tune these non-privately for each dataset, our implementation of DPSGD is also artificially strong.

All experiment code can be found on Github (Google, 2022).

## 4.2 DATASETS

We evaluate all four algorithms on the following datasets. The first dataset is synthetic, and the rest are real. The datasets are intentionally selected to be relatively easy use cases for linear regression, as reflected by the consistent high $R^2$ for NonDP.[2] However, we emphasize that, beyond the constraints on $d$ and $n$ suggested by Section 4.3, they have not been selected to favor TukeyEM: all feature selection occurred before running any of the algorithms, and we include all datasets evaluated where NonDP achieved a positive $R^2$. A complete description of the datasets appears both in the public code and the Appendix's Section 7.5. For each dataset, we additionally add an intercept feature.

1. **Synthetic** ($d = 11$, $n = 22{,}000$, Pedregosa et al. (2011)). This dataset uses `sklearn.make_regression` and $N(0, \sigma^2)$ label noise with $\sigma = 10$.

2. **California** ($d = 9$, $n = 20{,}433$, Nugent (2017)) predicting house price.

3. **Diamonds** ($d = 10$, $n = 53{,}940$, Agarwal (2017)), predicting diamond price.

4. **Traffic** ($d = 3$, $n = 7{,}909$, NYSDOT (2013)), predicting number of passenger vehicles.

5. **NBA** ($d = 6$, $n = 21{,}613$, Lauga (2022)), predicting home team score.

6. **Beijing** ($d = 25$, $n = 159{,}375$, ruiqurm (2018)), predicting house price.

7. **Garbage** ($d = 8$, $n = 18{,}810$, DSNY (2022)), predicting tons of garbage collected.

8. **MLB** ($d = 11$, $n = 140{,}657$, Samaniego (2018)), predicting home team score.

---

[2]$R^2$ measures the variation in labels accounted for by the features. $R^2 = 1$ is perfect, $R^2 = 0$ is the trivial baseline achieved by simply predicting the average label, and $R^2 < 0$ is worse than the trivial baseline.

### 4.3 CHOOSING THE NUMBER OF MODELS

Before turning to the results of this comparison, recall from Section 3 that TukeyEM privately aggregates $m$ non-private OLS models. If $m$ is too low, PTRCheck will probably fail; if $m$ is too high, and each model is trained on only a small number of points, even a non-private aggregation of inaccurate models will be an inaccurate model as well.

Experiments on synthetic data support this intuition. In the left plot in Figure 2, each solid line represents synthetic data with a different number of features, generated by the same process as the Synthetic dataset described in the previous section. We vary the number of models $m$ on the $x$-axis and plot the distance computed by Lemma 3.6. As $d$ grows, the number of models required to pass the PTRCheck threshold, demarcated by the dashed horizontal line, grows as well.

To select the value of $R^2$ used for TukeyEM, we ran it on each dataset using $m = 250, 500, \ldots, 2000$ and selected the smallest $m$ where all PTR checks passed. We give additional details in Section 7.6 but note here that the resulting choices closely track those given by Figure 2. Furthermore, across many datasets, simply selecting $m = 1000$ typically produces nearly optimal $R^2$, with several datasets exhibiting little dependence on the exact choice of $m$.

### 4.4 ACCURACY COMPARISON

Our main experiments compare the four methods at $(\ln(3), 10^{-5})$-DP. A concise summary of the experiment results appears in Figure 1. For every method other than NonDP (which is deterministic), we report the median $R^2$ values across the trials. For each dataset, the methods with interquartile ranges overlapping that of the method with the highest median $R^2$ are bolded. Extended plots recording $R^2$ for various $m$ appear in Section 7.6. All datasets use 10 trials, except for California and Diamonds, which use 50.

| Dataset | NonDP | AdaSSP | TukeyEM | DPSGD (tuned) | DPSGD (90% tuned) |
|---------|-------|--------|---------|---------------|-------------------|
| Synthetic | 0.997 | 0.991 | **0.997** | **0.997** | **0.997** |
| California | 0.637 | -1.285 | **0.099** | **0.085** | -1.03 |
| Diamonds | 0.907 | 0.216 | 0.307 | **0.828** | 0.371 |
| Traffic | 0.966 | 0.944 | **0.965** | 0.938 | 0.765 |
| NBA | 0.621 | 0.018 | **0.618** | 0.531 | 0.344 |
| Beijing | 0.702 | 0.209 | **0.698** | 0.475 | 0.302 |
| Garbage | 0.542 | 0.119 | **0.534** | 0.215 | 0.152 |
| MLB | 0.722 | 0.519 | **0.721** | 0.718 | 0.712 |

Figure 1: For each dataset, the DP methods with interquartile ranges overlapping that of the DP method with the highest median $R^2$ are bolded.

A few takeaways are immediate. First, on most datasets TukeyEM obtains $R^2$ exceeding or matching that of both AdaSSP and DPSGD. TukeyEM achieves this even though AdaSSP receives non-private access to the true feature and label norms, and DPSGD receives non-private access to extensive hyperparameter tuning.

We briefly elaborate on the latter. Our experiments tune DPSGD over a large grid consisting of 2,184 joint hyperparameter settings, over **learning_rate** $\in \{10^{-6}, 10^{-5}, \ldots, 1\}$, **clip_norm** $\in \{10^{-6}, 10^{-5}, \ldots, 10^{6}\}$, **microbatches** $\in \{2^5, 2^6, \ldots, 2^{10}\}$, and **epochs** $\in \{1, 5, 10, 20\}$. Ignoring the extensive computational resources required to do so at this scale (100 trials of each of the 2,184 hyperparameter combinations, for each dataset), we highlight that even mildly suboptimal hyperparameters are sufficient to significantly decrease DPSGD's utility. Figure 1 quantifies this by recording the $R^2$ obtained by the hyperparameters that achieved the highest and 90th percentile median $R^2$ during tuning. While the optimal hyperparameters consistently produce results competitive with or sometimes exceeding that of TukeyEM, even the mildly suboptimal hyperparameters nearly always produce results significantly worse than those of TukeyEM. The exact hyperparameters used appear in Section 7.6.

We conclude our discussion of DPSGD by noting that it has so far omitted any attempt at differentially private hyperparameter tuning. We suggest that the results here indicate that any such method will

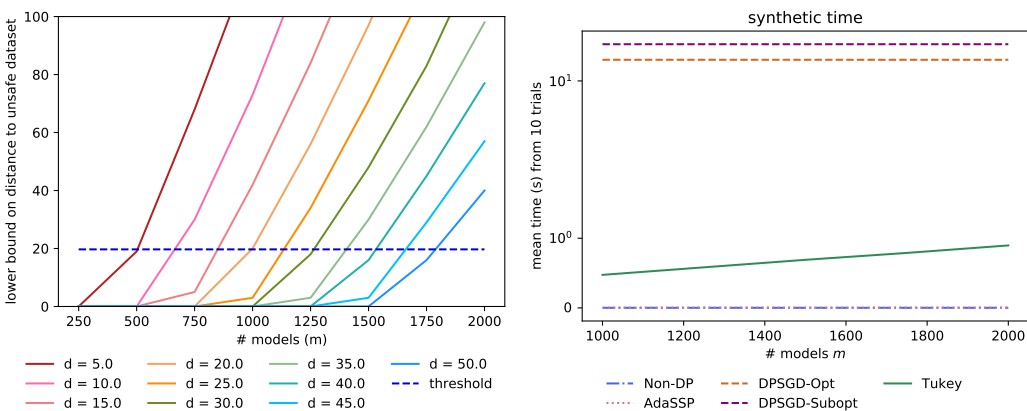

Figure 2: Left: plot of Hamming distance to unsafety using Lemma 3.6 as the feature dimension $d$ and number of models $m$ varies, using $(\ln(3), 10^{-5})$-DP and $n = (d+1)m$ throughout. Right: plot of average time in seconds as the number of models $m$ used by TukeyEM varies

need to select hyperparameters with high accuracy while using little privacy budget, and emphasize that the presentation of DPSGD in our experiments is generous.

Overall, TukeyEM's overall performance on the eight datasets is strong. We propose that the empirical evidence is enough to justify TukeyEM as a first-cut method for linear regression problems whose data dimensions satisfy its heuristic requirements ($n \gtrsim 1000 \cdot d$).

## 4.5 TIME COMPARISON

We conclude with a brief discussion of runtime. The rightmost plot in Figure 2 records the average runtime in seconds over 10 trials of each method. TukeyEM is slower than the covariance matrix-based methods NonDP and AdaSSP, but it still runs in under one second, and it is substantially faster than DPSGD. TukeyEM's runtime also, as expected, depends linearly on the number of models $m$. Since the plots are essentially identical across datasets, we only include results for the Synthetic dataset here. Finally, we note that, for most reasonable settings of $m$, TukeyEM has runtime asymptotically identical to that of NonDP (Theorem 1.1). The gap in practical performance is likely a consequence of the relatively unoptimized nature of our implementation.

## 5 FUTURE DIRECTIONS

An immediate natural extension of TukeyEM would generalize the approach to similar problems such as logistic regression. More broadly, while this work focused on linear regression for the sake of simplicity and wide applicability, the basic idea of TukeyEM can in principle be applied to select from arbitrary non-private models that admit expression as vectors in $\mathbb{R}^d$. Part of our analysis observed that TukeyEM may benefit from algorithms and data that lead to Gaussian-like distributions over models; describing the characteristics of algorithms and data that induce this property — or a similar property that better characterizes the performance of TukeyEM — is an open question.

## 6 ACKNOWLEDGMENTS

We thank Gavin Brown for helpful discussion of Brown et al. (2021), and we thank Jenny Gillenwater for useful feedback on an early draft. We also thank attendees of the Fields Institute Workshop on Differential Privacy and Statistical Data Analysis for helpful general discussions.

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

# 7 Appendix

## 7.1 Illustration of Exact and Approximate Tukey Depth

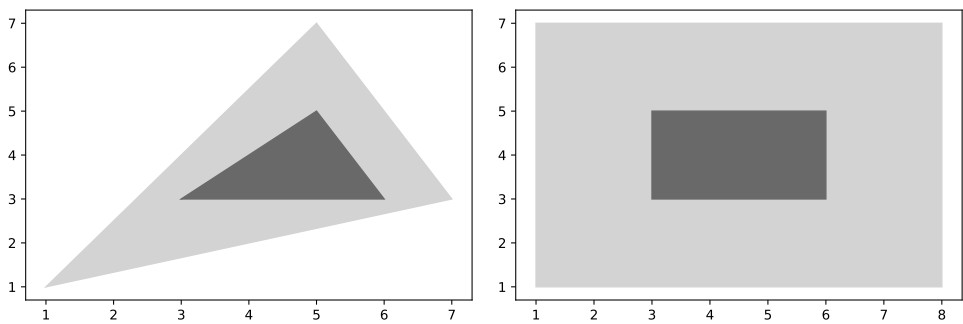

Figure 3: An illustrated comparison between exact (left) and approximate (right) Tukey depth. In both figures, the set of points is $\{(1,1),(7,3),(5,7),(3,3),(5,5),(6,3)\}$, the region of depth 0 is white, the region of depth 1 is light gray, and the region of depth 2 is dark gray. Note that for exact Tukey depth, the regions of different depths form a sequence of nested convex polygons; for approximate Tukey depth, they form a sequence of nested rectangles.

## 7.2 Omitted Proofs

We start with the proof of our utility result for exact Tukey depth, Theorem 3.1.

*Proof of Theorem 3.1.* This is a direct application of the results of Brown et al. (2021). They analyzed a notion of probabilistic (and normalized) Tukey depth over samples from a distribution: $T_{\mathcal{N}(\mu,\Sigma)}(y) := \min_v \mathbb{P}_{X \sim \mathcal{N}(\mu,\Sigma)}[\langle X, v \rangle \geq \langle y, v \rangle]$. Their Proposition 3.3 shows that $T_{\mathcal{N}(\mu,\Sigma)}(y)$ can be characterized in terms of $\Phi$, the CDF of the standard one-dimensional Gaussian distribution. Specifically, they show $T_{\mathcal{N}(\mu,\Sigma)}(y) = \Phi(-\|y-\mu\|_\Sigma)$. From their Lemma 3.5, if $m \geq c \left( \frac{d + \log(1/\gamma)}{\alpha^2} \right)$, then with probability $1 - \gamma$, $|p/m - T_{\mathcal{N}(\beta^*,\Sigma)}(\hat{\beta})| \leq \alpha$. Thus

$$-\alpha \leq T_{\mathcal{N}(\beta^*,\Sigma)}(\hat{\beta}) - p/m$$
$$p/m - \alpha \leq \Phi(-\|\hat{\beta} - \beta^*\|_\Sigma)$$
$$p/m - \alpha \leq 1 - \Phi(\|\hat{\beta} - \beta^*\|_\Sigma)$$
$$\Phi(\|\hat{\beta} - \beta^*\|_\Sigma) \leq 1 - p/m + \alpha$$
$$\|\hat{\beta} - \beta^*\|_\Sigma \leq \Phi^{-1}(1 - p/m + \alpha)$$

where the third inequality used the symmetry of $\Phi$. $\qquad\square$

Next, we prove our result about computing the volumes associated with different approximate Tukey depths, Lemma 3.3.

*Proof of Lemma 3.3.* By the definition of approximate Tukey depth, for arbitrary $y = (y_1, \ldots, y_d)$ of Tukey depth at least $i$, each of the $2d$ halfspaces $h_{y_1 \cdot e_1}, h_{y_1 \cdot -e_1}, \ldots, h_{y_d \cdot e_d}, h_{y_d \cdot -e_d}$ contains at least $i$ points from $D$, where $x \cdot y$ denotes multiplication of a scalar and vector. Fix some dimension $j \in [d]$. Since $\min(|h_{y_j \cdot e_j} \cap D|, |h_{y_j \cdot -e_j} \cap D|) \geq i$, $y_j \in [S_{j,i}, S_{j,m-(i-1)}]$. Thus $V_{i,D} = \prod_{j=1}^d (S_{j,m-(i-1)} - S_{j,i})$. The computation of $S$ starting in Line 5 sorts $d$ arrays of length $m$ and so takes time $O(dm \log(m))$. Line 9 iterates over $m/2$ depths and computes $d$ quantities, each in constant time, so its total time is $O(dm)$. $\qquad\square$

The next result is our 1-sensitive and efficient adaptation of the lower bound from Brown et al. (2021). We first restate that result. While their paper uses swap DP, the same result holds for add-remove DP.

**Lemma 7.1** (Lemma 3.8 (Brown et al., 2021)). *For any $k \geq 0$, if there exists a $g > 0$ such that $\frac{V_{t-k-1,D}}{V_{t+k+g+1,D}} \cdot e^{-\varepsilon g/2} \leq \delta$, then for every database $z$ in $\mathtt{Unsafe}_{(\varepsilon, 4e^\varepsilon \delta, t)}$, $d_H(D, z) > k$, where $d_H$ denotes Hamming distance.*

We now prove its adaptation, Lemma 3.6.

*Proof of Lemma 3.6.* We first prove item 1. Let $D$ and $D'$ be neighboring databases, $D' = D \cup \{x\}$, and let $k_D^*$ and $k_{D'}^*$ denote the mechanism's outputs on the respective databases. It suffices to show $|k_D^* - k_{D'}^*| \leq 1$.

Consider some $V_p$ for nonnegative integer $p$. If $x$ has depth less than $p$ in $D$, then $V_{p-1,D} \geq V_{p,D'} \geq V_{p,D} > V_{p+1,D}$. Otherwise, $V_{p+1,D} < V_{p,D} = V_{p,D'} < V_{p-1,D}$. In either case,

$$V_{p+1,D} < V_{p,D} \leq V_{p,D'} \leq V_{p-1,D}. \tag{1}$$

Now suppose there exist $k_{D'}^* \geq 0$ and $g_{D'}^* > 0$ such that $\frac{V_{t-k_{D'}^*-1,D'}}{V_{t+k_{D'}^*+g_{D'}^*+1,D'}} \cdot e^{-\varepsilon g_{D'}^*/2} \leq \delta$. Then by Equation 1, $\frac{V_{t-k_{D'}^*,D}}{V_{t+k_{D'}^*+g_{D'}^*,D}} \cdot e^{-\varepsilon g_{D'}^*/2} \leq \delta$, so $k_D^* \geq k_{D'}^* - 1$. Similarly, if there exist $k_D^* \geq 0$ and $g_D^* > 0$ such that $\frac{V_{t-k_D^*-1,D}}{V_{t+k_D^*+g_D^*+1,D}} \cdot e^{-\varepsilon g_D^*/2} \leq \delta$, then by Equation 1, $\frac{V_{t-k_D^*,D'}}{V_{t+k_D^*+g_D^*,D'}} \cdot e^{-\varepsilon g_{D'}^*/2} \leq \delta$, so $k_{D'}^* \geq k_D^* - 1$. Thus if $k_D^* \geq 0$ or $k_{D'}^* \geq 0$, $|k_D^* - k_{D'}^*| \leq 1$. The result then follows since $k^* \geq -1$.

We now prove item 2. This holds for $k \geq 0$ by Lemma 7.1; for $k = -1$, the lower bound on distance to unsafety is the trivial one, $d_H(D, z) \geq 0$. $\square$

The next proof is for Lemma 3.7, which verifies the overall privacy and runtime of PTRCheck.

*Proof of Lemma 3.7.* The privacy guarantee follows from the 1-sensitivity of computing $k$ (Lemma 3.6). For the runtime guarantee, we perform a binary search over the distance lower bound $k$ starting from the maximum possible $m/4$ and, for each $k$, an exhaustive search over $g$. Note that if some $k, g$ pair satisfies the inequality in Lemma 7.1, there exists some $g'$ for every $k' < k$ that satisfies it as well. Thus since both have range $\leq m/4$, the total time is $O(m \log(m))$. $\square$

### 7.3 USING MONOTONICITY

This section discusses our use of monotonicity in the restricted exponential mechanism. Definition 2.2 states that, if $u$ is monotonic, the exponential mechanism can sample an output $y$ with probability proportional to $\exp\left(\frac{\varepsilon u(D,y)}{\Delta_u}\right)$ and satisfy $\varepsilon$-DP. Approximate Tukey depth is monotonic, so our application can also sample from this distribution. It remains to incorporate monotonicity into the PTR step.

It suffices to show that Lemma 7.1 also holds for a restricted exponential mechanism using a monotonic score function. Turning to the proof of Lemma 7.1 given by Brown et al. (2021), it suffices to prove their Lemma 3.7 using $w_x(S) = \int_S \exp(\varepsilon q(x; y)) dy$. Note that their $w_x(S)$ differs by the 2 in its denominator inside the exponent term; this modification is where we incorporate monotonicity. This difference shows up in two places in their argument. First, we can replace their bound

$$\frac{\mathbb{P}[M_{\varepsilon,t}(x) = y]}{\mathbb{P}[M_{\varepsilon,t}(x') = y]} \leq e^{\varepsilon/2} \cdot \frac{w_{x'}(Y_{t,x'})}{w_x(Y_{t,x})} \leq e^\varepsilon \cdot \frac{w_x(Y_{t,x'})}{w_x(Y_{t,x})}$$

with the two cases that arise in add-remove differential privacy. The first considers $x' \subsetneq x$ and yields

$$\frac{\mathbb{P}[M_{\varepsilon,t}(x) = y]}{\mathbb{P}[M_{\varepsilon,t}(x') = y]} \leq e^\varepsilon \cdot \frac{w_{x'}(Y_{t,x'})}{w_x(Y_{t,x})} \leq e^\varepsilon \cdot \frac{w_x(Y_{t,x'})}{w_x(Y_{t,x})}$$

since the mechanism on $x$ never assigns a lower score to an output than on $x'$. Using the same logic, the second considers $x \subsetneq x'$, and we get

$$\frac{\mathbb{P}[M_{\varepsilon,t}(x) = y]}{\mathbb{P}[M_{\varepsilon,t}(x') = y]} \leq \frac{w_{x'}(Y_{t,x'})}{w_x(Y_{t,x})} \leq e^\varepsilon \cdot \frac{w_x(Y_{t,x'})}{w_x(Y_{t,x})}.$$

The second application in their argument, which bounds $\frac{\mathbb{P}[M_{\varepsilon,t}(x')=y]}{\mathbb{P}[M_{\varepsilon,t}(x)=y]}$, uses the same logic. As a result, their Lemma 3.7 also holds for a monotonic restricted exponential mechanism, and we can drop the 2 in the sampling distribution as desired.

## 7.4 SAMPLING FROM A REGION DETAILS

We start by formally defining our partition.

**Definition 7.2.** *Given $d$-dimensional database $D$ and dimension $j \in [d]$, for $y \in \mathbb{R}^d$, let $T_{D,j}(y)$ denote the exact (one-dimensional) Tukey depth of point $y$ with respect to dimension $j$ in database $D$. Let $B_i$ denote the region of points with approximate Tukey depth $i$. Define the partition $\{C_{j,i}\}_{j=1}^d$ of $B_i$ as the volume of points where depth $i$ occurs in dimension $j$ for the first time, i.e.,*

$$C_{j,i} := \{y \in \mathbb{R}^d \mid \min_{j'<j} T_{D,j'}(y) > i \text{ and } T_{D,j}(y) = i \text{ and } \min_{j'>j} T_{D,j'}(y) \geq i\}.$$

The partition is well defined because any point with approximate Tukey depth $i$ is in exactly one of the $C_{j,i}$ volumes. Each $C_{j,i}$ is also the Cartesian product of three sets: any $y \in C_{j,i}$ must have 1) depth strictly greater than $i$ in dimensions $1, ..., j-1$, 2) depth $i$ in dimension $j$, and 3) depth at least $i$ in dimensions $j+1, ..., d$. Being the Cartesian product of three sets, the total volume of $C_{j,i}$ can be computed as the product of the three corresponding volumes in lower dimensions. We will denote these by $V_{<j,i}, W_{j,i}, V_{>j,i}$, formalized below.

**Definition 7.3.** *Given $d$-dimensional database $D$, dimension $j \in [d]$, and depth $i$, define*

1. *$V_{j,i,D} = \mathsf{vol}(\{y_j \mid y \in \mathbb{R}^d, \tilde{T}_D(y) \geq i\})$, the total length in dimension $j$ of the region with approximate Tukey depth at least $i$.*

2. *$W_{j,i,D} = \mathsf{vol}(\{y_j \mid y \in \mathbb{R}^d, T_{D,j}(y) = i \text{ and } \tilde{T}_D(y) \geq i\})$, the total length in dimension $j$ of the region with depth exactly $i$ in dimension $j$ and approximate Tukey depth at least $i$.*

3. *$V_{<j,i,D} = \mathsf{vol}(\{y_{1:j-1} \mid y \in \mathbb{R}^d, \tilde{T}_D(y) \geq i\})$, the volume of the projection onto the first $j-1$ dimensions of points with approximate Tukey depth at least $i$. Define $V_{>j,i,D}$ analogously.*

*When $D$ is clear from context, we drop it from the subscript.*

The next lemma shows how to compute these and other relevant volumes. We again note that we fix $m$ to be even for neatness. The odd case is similar.

**Lemma 7.4.** *Given matrix $S \in \mathbb{R}^{d \times m}$ of projected and sorted models, as in Line 8 of Algorithm 2,*

1. *$V_{j,i} = S_{j,m-(i-1)} - S_{j,i}$,*

2. *$W_{j,i} = V_{j,i} - V_{j,i+1}$,*

3. *$V_{<j,i} = \prod_{j'=1}^{j-1} V_{j,i}$ and $V_{>j,i} = \prod_{j'=j+1}^{d} V_{j,i}$*

*Proof.* The proofs of the first item uses essentially the same reasoning as the proof of Lemma 3.3. For the second item, any point contributing to $V_{j,i}$ but not $V_{j,i+1}$ has depth exactly $i$ in dimension $j$. For the third item, a point contributes to $V_{<j,i}$ if and only if it has depth at least $i$ in all $d$ dimensions; since the resulting region is a rectangle, its volume is the product of its side lengths. □

With Lemma 7.4, we can now prove that SamplePointWithDepth works as intended.

*Proof of Lemma 3.9.* Given $S \in \mathbb{R}^{d \times m}$, define $B_i$ and $\{C_{j,i}\}_{j=1}^d$ as in Definition 7.2. To show the outcome of SamplePointWithDepth$(S, i)$ is uniformly distributed over points with approximate Tukey depth $i$, it suffices to show the algorithm samples a $C_{j,i}$ with probability proportional to its volume. Recall that $C_{j,i}$ is the set of points with depth greater than $i$ in dimensions $1, 2, \ldots, j-1$, exactly $i$ in dimension $j$, and at least $i$ in the remaining dimensions. With this interpretation, $C_{j,i}$ is

---

**Algorithm 3** RestrictedTukeyEM

---

1: **Input:** Tukey depth region volumes $V$, sorted collection of estimators $S$, depth restriction $t$, privacy parameter $\varepsilon$
2: **for** $i = t, t+1, \ldots, |V| - 1$ **do**
3:     Compute volume of region of Tukey depth exactly $i$, $W_i \leftarrow V_i - V_{i+1}$
4: Sample depth $\hat{i}$ from distribution where $\mathbb{P}\left[i\right] \propto W_i \exp\left(\frac{\varepsilon \cdot i}{2}\right)$
5: Return $y \leftarrow \mathsf{SamplePointWithDepth}(S, \hat{i})$

---

**Algorithm 4** SamplePointWithDepth

---

1: **Input:** Sorted collection of estimators $S$, depth $i$
2: $d, m \leftarrow$ number of rows and columns in $S$
3: Compute $V_{\geq 1, i}$ using Lemma 7.4
4: **for** $j = 1, \ldots, d$ **do**
5:     Compute $V_{<j, i+1}$, $W_{j,i}$, and $V_{>j,i}$ using Lemma 7.4
6:     Compute $\mathsf{vol}(C_{j,i}) = V_{<j,i+1} \cdot W_{j,i} \cdot V_{>j,i}$
7: Sample index $j^* \in [d]$ with probability $\frac{\mathsf{vol}(C_{j,i})}{V_{\geq 1, i}}$
8: **for** $j' = 1, \ldots, j^* - 1$ **do**
9:     $y_{j'} \leftarrow$ uniform random sample from $[S_{j', i+1}, S_{j', m-i}]$
10: $y_{j^*} \leftarrow$ uniform random sample from $[S_{j^*, i}, S_{j^*, i+1}) \cup (S_{j^*, m-i}, S_{j^*, m-(i-1)}]$
11: **for** $j' = j^* + 1, \ldots, d$ **do**
12:     $y_{j'} \leftarrow$ uniform random sample from $[S_{j', i}, S_{j', m-(i-1)}]$
13: Return $y$

---

the Cartesian product of three lower dimensional regions, and thus its volume is the product of the corresponding volumes,

$$\mathsf{vol}(C_{j,i}) = V_{<j,i+1} \cdot W_{j,i} \cdot V_{>j,i}.$$

These quantities, along with the normalizing constant $V_{\geq 1, i}$, can be computed using Lemma 7.4.

Since $i$ is fixed, computing the full set of $V_{j,i}$ and $W_{j,i}$ takes time $O(d)$, and by tracking partial sums and using logarithms, we can compute the full set of $V_{<j,i}$ and $V_{>j,i}$ in time $O(d)$ as well. The last step is sampling the final point $y$, which takes time $O(d)$ using the previously computed $S$. □

## 7.5 DATASET FEATURE SELECTION DETAILS

This section provides details for each of the real datasets evaluated in our experiments.

1. **California Housing** Nugent (2017): The label is `median_housevalue`, and the categorical `ocean_proximity` is dropped.

2. **Diamonds** Agarwal (2017): The label is `price`. Ordinal categorical features (`carat`, `color`, `clarity`) are replaced with integers $1, 2, \ldots$.

3. **Traffic** NYSDOT (2013): The label is passenger vehicle count (`Class 2`), and the remaining features are motorcycles (`Class 1`) and pickups, panels, and vans (`Class 3`).

4. **NBA** Lauga (2022): The label is `PTS_home`, and the features are `FT_PCT_home`, `FG3_PCT_home`, `FG_PCT_home`, `AST_home`, and `REB_home`.

5. **Beijing Housing** ruiqurm (2018): The label is `totalPrice`. Features are days on market (`DOM`), `followers`, area of house in meters (`square`), number of kitchens (`kitchen`), `buildingType`, `renovationCondition`, building material (`buildingStructure`), ladders per residence (`ladderRatio`), elevator presence `elevator`, whether previous owner owned for at least five years (`fiveYearsProperty`), proximity to subway (`subway`), `district`, and nearby average housing price (`communityAverage`). Categorical `buildingType`, `renovationCondition`, and `buildingStructure` are encoded as one-hot variables. We additionally removed a single outlier row (60422) whose norm is more than two

orders of magnitude larger than that of other points; none of the DP algorithms achieved positive $R^2$ with the row included.

6. **New York Garbage** DSNY (2022): The label is REFUSETONSCOLLECTED. The features are PAPERTONSCOLLECTED and MPGTONSCOLLECTED. The categorical BOROUGH is encoded as one-hot variables.

7. **MLB** Samaniego (2018): The label is home team runs (h_score) and the features are v_strikeouts, v_walks, v_pitchers_used, v_errors, h_at_bats, h_hits, h_doubles, h_triples, h_homeruns, and h_stolen_bases.

## 7.6 EXTENDED EXPERIMENT RESULTS

Figure 4 records, respectively, the number of epochs, the clipping norm, the learning rate, and the number of microbatches for both the best and 90th percentile hyperparameter settings on each dataset.

| Dataset | DPSGD (tuned) | DPSGD (90% tuned) |
|---|---|---|
| Synthetic | $(20, 1, 1, 128)$ | $(20, 10^{-3}, 0.1, 64)$ |
| California | $(20, 100, 1, 64)$ | $(10, 1000, 0.01, 64)$ |
| Diamonds | $(20, 10^6, 1, 128)$ | $(10, 10^6, 0.1, 32)$ |
| Traffic | $(1, 10^6, 1, 1024)$ | $(10, 10^5, 0.1, 512)$ |
| NBA | $(20, 100, 1, 512)$ | $(20, 10^{-6}, 0.1, 32)$ |
| Beijing | $(20, 100, 0.01, 512)$ | $(20, 0.1, 0.001, 512)$ |
| Garbage | $(20, 1, 1, 32)$ | $(5, 1000, 1, 64)$ |
| MLB | $(10, 100, 0.01, 512)$ | $(10, 10^{-5}, 0.001, 128)$ |

Figure 4: Hyperparameter settings used by DPSGD on each dataset.

## 7.7 DISTRIBUTION OF MODELS FOR ALL DATASETS

Figures 7 through 13 display histograms of models trained on each dataset. For each plot, we train 2,000 standard OLS models on disjoint partitions of the data and plot the resulting histograms for each coefficient. The red curve plots a Gaussian distribution with the same mean and standard deviation as the underlying data.

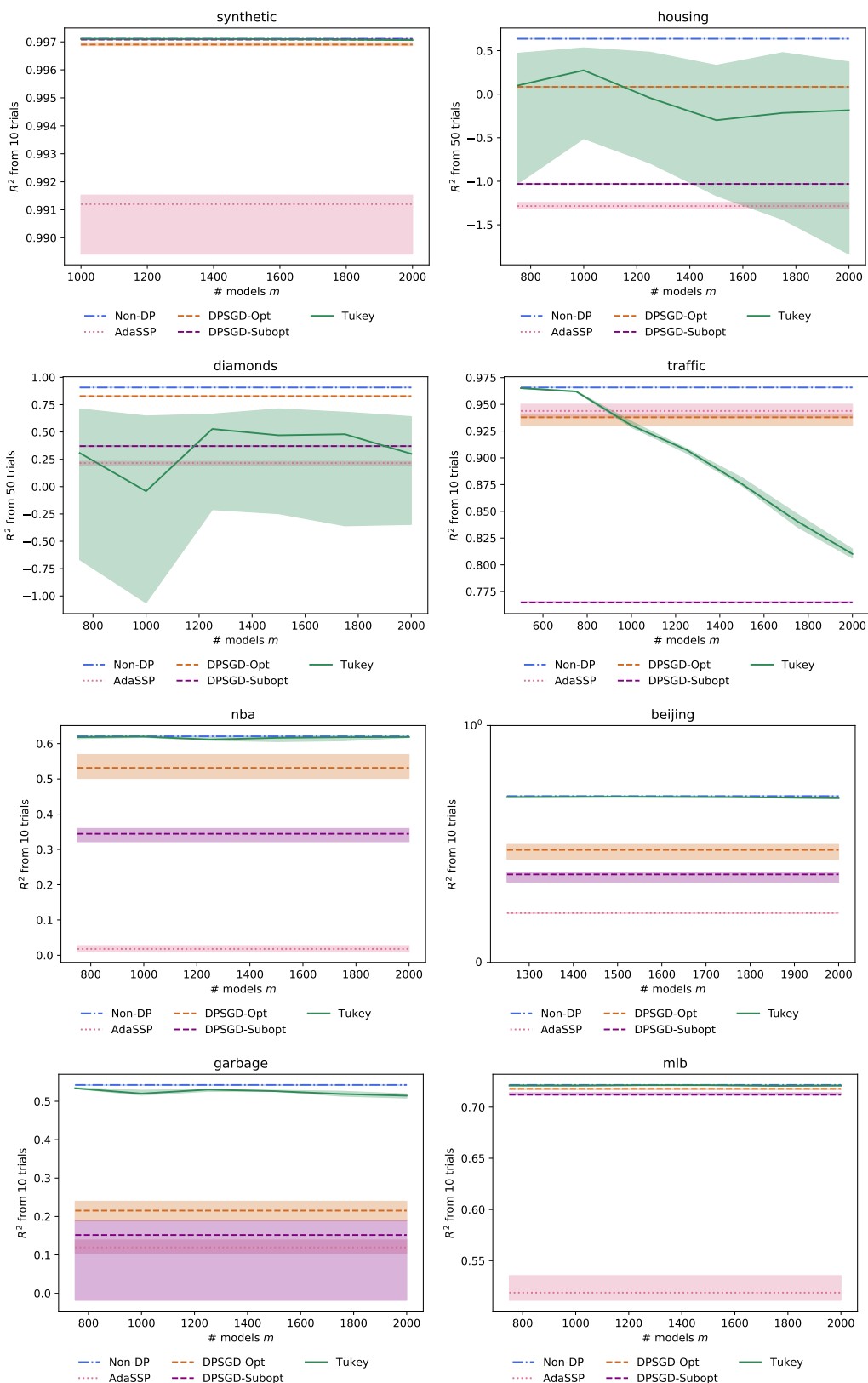

Figure 5: Plots of $R^2$ as the number of models $m$ used by TukeyEM varies. The lines mark medians and the shaded regions span the first and third quartiles. All datasets except Housing and Diamonds use 10 trials. Housing and Diamonds use 50 trials due to the variance of TukeyEM. Methods other than TukeyEM appear as flat lines because they do not vary with $m$. Each plot varies the number of models $m$ in increments of 250, starting with the $m$ sufficient to pass PTR in all trials.

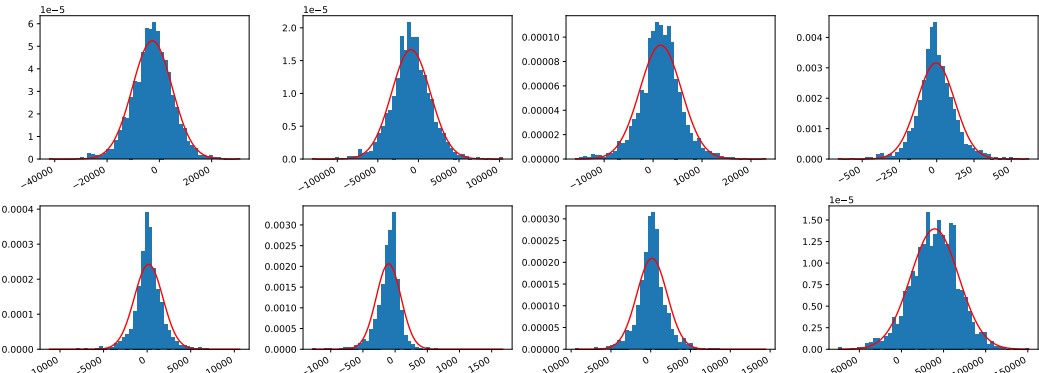

Figure 6: Histograms of models on the California dataset.

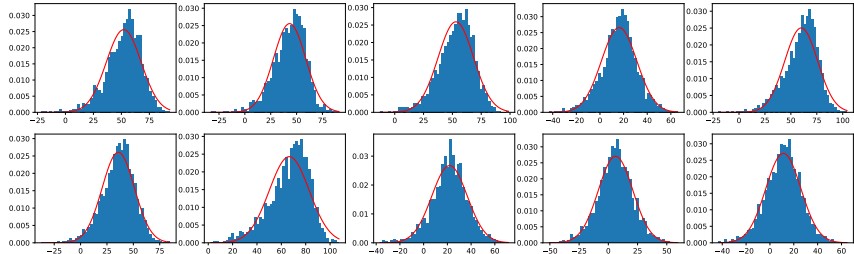

Figure 7: Histograms of models on Synthetic.

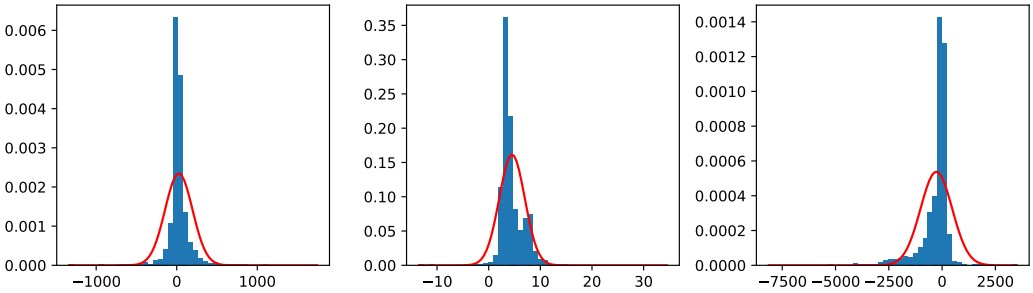

Figure 8: Histograms of models on Traffic.

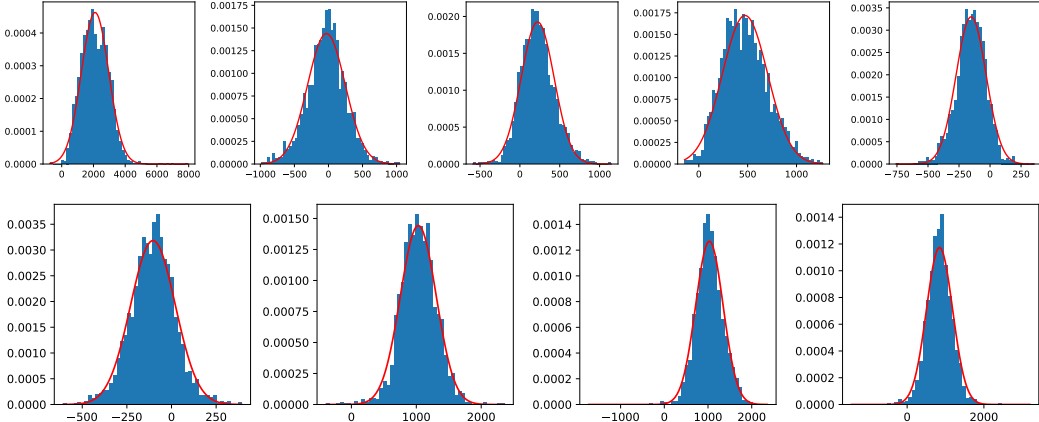

Figure 9: Histograms of models on Diamonds.

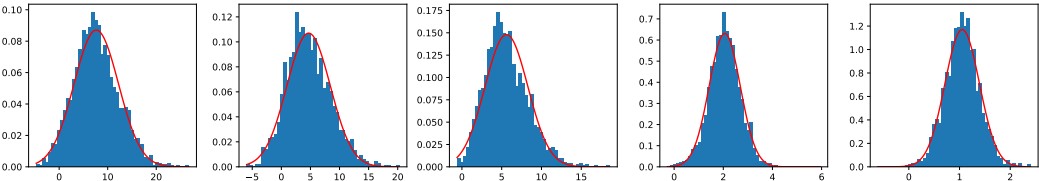

Figure 10: Histograms of models on NBA.

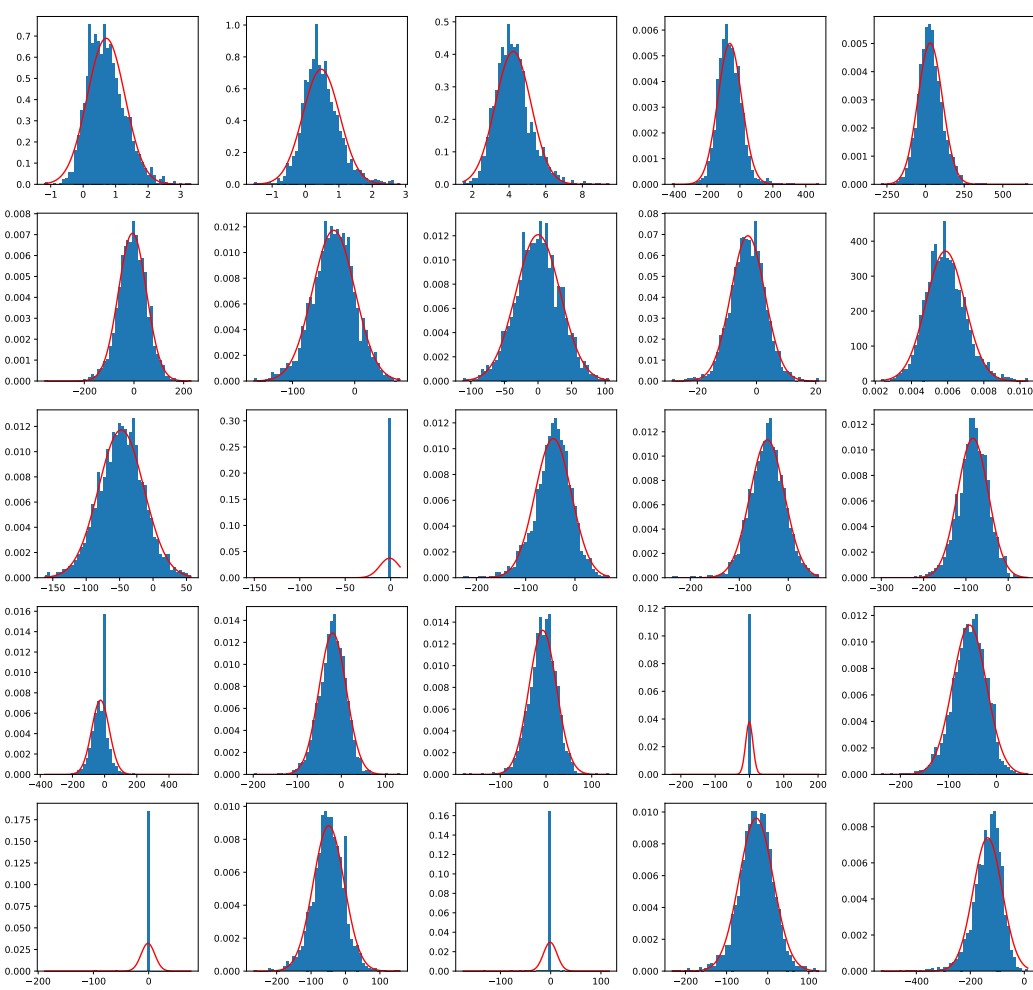

Figure 11: Histograms of models on Beijing.

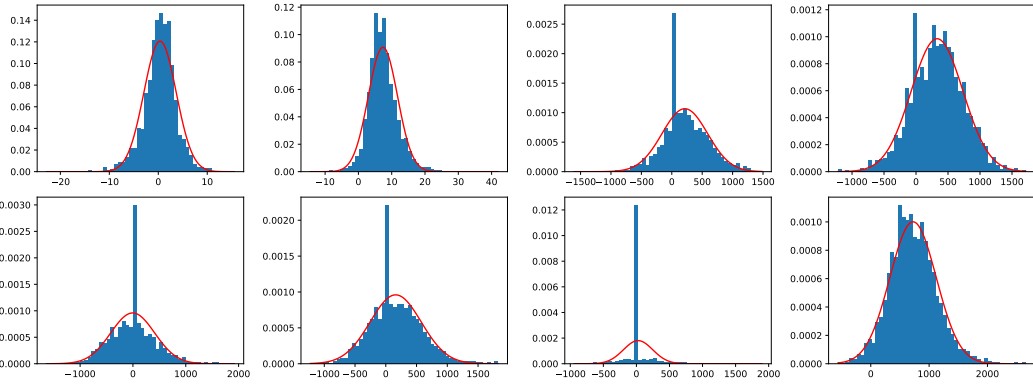

Figure 12: Histograms of models on Garbage.

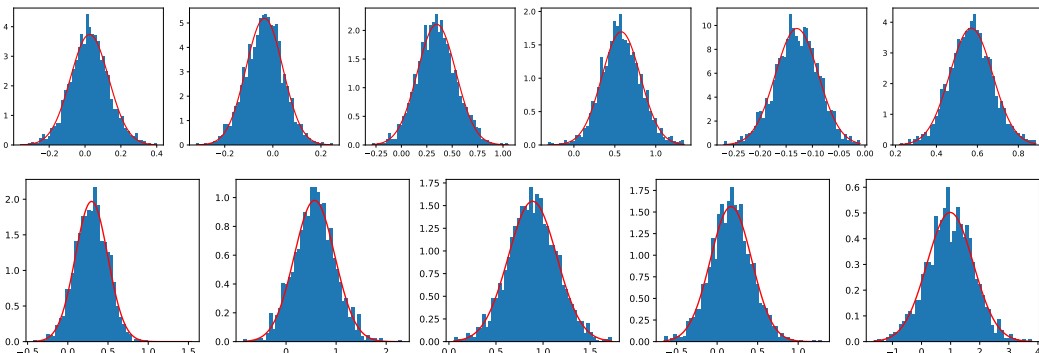

Figure 13: Histograms of models on MLB.