# OpenReview forum: "Easy Differentially Private Linear Regression"
_ICLR.cc/2023/Conference — ICLR 2023 poster_

### Official Review · Reviewer_3oJ7 · 2022-10-23

**Confidence:** 3
**Correctness:** 3
**Technical Novelty And Significance:** 3
**Empirical Novelty And Significance:** 3
**Recommendation:** 6

**Clarity, Quality, Novelty And Reproducibility:**

This paper proposes a method for private linear regression setting. Their Tukey depth-based method generalizes the previous work and allows to work with datasets with larger feature size than the existing methods. Paper is well-designed and highly detailed as the authors provide proofs and details elaboratively throughout the paper. Also, they discuss several possible drawbacks of their method extensively.

**Strength And Weaknesses:**

Strengths:

The proposed method  is parameter-free and it does not need for pre distributional assumptions.. TukeyEM performs well compared against other competing models although some of them are allowed to have non-private information (for ex. DP-SGD has access to the non-private parameter tuning, adaSSP has access to the non-private data bounds).

Weaknesses:

1. The method needs a pre-defined m (number of subsets of to divide whole data), and this may be problematic as small m leads failures in PTR check, and larger m means not enough data to learn in step 1 of the algorithm (this issue is also partly acknowledged by the authors). They provide a method for choosing m in Section 4.3 (run the models with various m's and choose the smallest one that leads all pass in PTR check). However is this a part of the main algorithm? Otherwise, how does an end-user decide on m without a detailed understanding of the method? The authors state that m=1000 is enough for most of the cases but it is hard to assess whether it is guaranteed for all real-life scenarios. Indeed, in some cases, number of models (m) may be large (such as 1250 as in Beijing data), and as they note this may result in poor performance for TukeyEM. Hence, this may lead the reader to the fact that TukeyEM works well when the data size is large, indeed they state n>=1000d. Can this equation be relaxed? That is, can TukeyEM work well with small datasets?

2. The authors present Theorem 3.1, which works with exact Tukey depth. But their algorithm is based on approximate Tukey depth. I could not assess what are the consequences of using approximate Tukey depth on the analysis?

3. The paragraph after Theorem 3.1 ends with the statement "[...] nor does non-Gaussianity preclude accurate estimation." Can you please elaborate as Gaussianity assumption is already in Theorem 3.1?

4. In Algorithm 2, line 2 the authors partition X not only "randomly," but also "evenly." Is there a particular reason for even partitioning?

5. How about the prediction performance of TukeyEM? If I am not mistaken, the authors have not provided any discussion on this part.

6. In the first step of the main algorithm, the subsets are partitioned randomly. Would it help to consider these subsets consisting of points that are close to each other (something similar to a nearest neighbour calculation)?

7. The Turkey dept can be written as an optimization problem with binary variables. Thus, when dataset size is reasonable (~10000), an exact optimal solution can be obtained. Then, this exact optimal can be compared against the approximate Turkey depth to empirically discuss the performance of the approximate algorithm. This is just a suggestion for the authors, and I do not expect them to do this for this submission.

**Summary Of The Paper:**

The authors have developed a (non-parametric) algorithm for Differentially Private Linear Regression problem. The model works without hyperparameters and privacy bounds using "Tukey depth". They call this method as TukeyEM. TukeyEM works in four steps: (i) splitting data into subsets, (ii) computing columns using approximate Turkey depths of given OLS estimators, (iii) adding noise after finding the maximum of the lower-bound, (iv) sampling solutions. With several numerical experiments using real-datasets, they show that TukeyEM performs well compared against several state-of-the-art methods, and it is computationally efficient.

**Summary Of The Review:**

A well-written paper on a novel differentially private linear regression. The authors have provided a detailed analysis along with a sufficient numerical experiments.

---

> ### Author Response · Authors · 2022-11-10
> **Response to reviewer 3oJ7**
>
> Thanks for the detailed review. We've highlighted questions/comments and attempted to answer them below. Please let us know if you have further questions.
>
> > … how does an end-user decide on m without a detailed understanding of the method? The authors state that m=1000 is enough for most of the cases but it is hard to assess whether it is guaranteed for all real-life scenarios. Indeed, in some cases, number of models (m) may be large (such as 1250 as in Beijing data), and as they note this may result in poor performance for TukeyEM. Hence, this may lead the reader to the fact that TukeyEM works well when the data size is large, indeed they state n>=1000d. Can this equation be relaxed? That is, can TukeyEM work well with small datasets?
>
> TukeyEM does work best when the data size is large. We note that even in this data-rich setting, TukeyEM is the only algorithm that reliably works well without either detailed knowledge of the data or extensive (and non-private) hyperparameter tuning. To the best of our understanding, relaxing the $n \geq 1000d$ heuristic would require a tighter analysis of the PTR step. We are not aware of obvious slack in the existing argument; optimizing this may be a reasonable direction for future work.
>
> > The authors present Theorem 3.1, which works with exact Tukey depth. But their algorithm is based on approximate Tukey depth. I could not assess what are the consequences of using approximate Tukey depth on the analysis?
>
> In the worst case there can be an arbitrary gap between exact Tukey depth and our approximation. Consider points in $\mathbb{R}^2$ which sit along a one dimensional subspace. For concreteness, say all the points sit on the line $y = x$. Let $m$ be the 1-D median determined by computing an ordinary median along the 1-D subspace. Note that if we perturb $m$ by adding a random epsilon-normed vector to it, we arrive at a point $m’$ that has a zero exact Tukey depth (w.h.p.), but still has large approximate Tukey depth.
>
> However, observe that his gap can be considered a feature of the approximation. Intuitively, $m’$ is a good model in that it is epsilon-close to a central model $m$. The fact that approximate Tukey depth is sometimes more robust to low rank data is perhaps surprising and indicates the need for novel theory to describe the worst-case behavior of the approximation. Finally, we emphasize that the quality of approximate Tukey depth's approximation of exact Tukey depth does not affect the privacy guarantee.
>
> > The paragraph after Theorem 3.1 ends with the statement "[...] nor does non-Gaussianity preclude accurate estimation." Can you please elaborate as Gaussianity assumption is already in Theorem 3.1?
>
> Theorem 3.1 uses Gaussianity as a sufficient condition for accurate estimation. The referenced comment is intended to point out that this does not mean Gaussianity is a necessary condition.
>
> > In Algorithm 2, line 2 the authors partition X not only "randomly," but also "evenly." Is there a particular reason for even partitioning?
>
> The random partition is symmetric – there is no a priori reason to believe that one partition is more or less useful than another – so partitioning evenly also seemed reasonable.
>
> > How about the prediction performance of TukeyEM? If I am not mistaken, the authors have not provided any discussion on this part.
>
> Correct, we did not report train/test splits in our experiments, largely because we were not concerned with the threat of overfitting using linear models. Re-running TukeyEM on each dataset for the same number of trials and the same number of models with a random 90-10 train-test split produced the following median test R^2 values. We also report the re-run values for AdaSSP in parentheses. Re-running DPSGD requires more time; we can attempt to provide results later in the response period if desired.
>
> Synthetic: 0.997 (0.992)
>
> California: -0.72 (-1.32)
>
> Diamonds: 0.20 (0.307)
>
> Traffic: 0.959 (0.937)
>
> NBA: 0.617 ( 0.007)
>
> Beijing: 0.696 (0.211)
>
> Garbage: 0.521 (0.109)
>
> MLB: 0.720 (0.542)
>
> We briefly note that the R^2 reported for Beijing is much higher here than in the submission. This is because while generating the results above we noticed large differences between train and test results for Beijing. We eventually found that there is a single outlier in the Beijing dataset (row 60422) whose norm is more than two orders of magnitude larger than other points, and this outlier skews R^2. Removing this outlier produced the much higher R^2 given above. (Note also that it significantly improved AdaSSP's R^2 as well, from -2.3 to 0.2.)
>
> (OpenReview prohibits long posts, so see the next post in this thread for further responses.)

---

> > ### Author Response · Authors · 2022-11-10
> > **Response to reviewer 3oJ7, continued**
> >
> > > In the first step of the main algorithm, the subsets are partitioned randomly. Would it help to consider these subsets consisting of points that are close to each other (something similar to a nearest neighbour calculation)?
> >
> > Note that in general non-random partitioning necessitates additional work in the privacy analysis to bound the contribution of a single point, since the addition or removal of one point may change both the partition it falls into as well as the content of other partitions. This makes a nearest neighbor-style clustering step difficult from a privacy perspective. However, if we ignore this issue, this step may also lead to highly inaccurate outputs. For example, suppose our dataset consists of $k$ clusters: one around $(1,1)$, another around $(2,2)$, …, and one around $(k, k)$. The data overall is clearly well fit by a linear model $y = x$, but training a separate model on each cluster will produce an essentially random direction.
> >
> > > The Turkey dept can be written as an optimization problem with binary variables. Thus, when dataset size is reasonable (~10000), an exact optimal solution can be obtained. Then, this exact optimal can be compared against the approximate Turkey depth to empirically discuss the performance of the approximate algorithm. This is just a suggestion for the authors, and I do not expect them to do this for this submission.
> >
> > Note that even if we assume access to an efficient oracle for computing exact Tukey depth, regions of different depths are polytopes for exact Tukey depth. Regions of different depths are rectangles for approximate Tukey depth. This matters because computing the volumes of different depths is a necessary step for sampling from the restricted exponential mechanism we use. To the best of our understanding, computing polytope volumes exactly – as the privacy analysis requires – is significantly more computationally intensive than computing rectangle volumes exactly. As a result, even access to an oracle for exact Tukey depth does not completely resolve the computational problems associated with exact Tukey depth.

---

### Official Review · Reviewer_Ndnv · 2022-10-24

**Confidence:** 2
**Correctness:** 4
**Technical Novelty And Significance:** 3
**Empirical Novelty And Significance:** 3
**Recommendation:** 8

**Clarity, Quality, Novelty And Reproducibility:**

The paper is well written. The problem is interesting. The algorithm seems to be novel.


**Details Of Ethics Concerns:**

No concerns.


**Strength And Weaknesses:**

Strengths:
(1) The motivation and goal are clear and compelling.
(2) The algorithm is indeed easy to use for the end-user. The only remaining trouble is the need for this parameter m.
(3) The paper is generally well written, and gives the reader intuition along with technical definitions and theorems.
(4) The experimental evaluation shows that the algorithm produces, for many data sets, good models (that are comparable to the non-DP baseline) in a reasonable amount of time.

Weakness:
(1) Specifying m may be difficult for an end user. (although, the authors give a heuristic for how one might choose m that seems to work well in their experiments)
(2) The possibility that the algorithm fails to return a model for some reasonable specifications of m seems annoying.
(3) The time complexity suggests that the method is not well suited for high-dimensional data.


**Summary Of The Paper:**

This paper presents a (eps, delta)-differentially private linear regression algorithm using the concept of Tukey depth. A primary goal of the paper is to reduce the amount of effort required of the end-user. In particular, the algorithm here simply requires the user to input the dataset, specify eps and delta, and additionally input a single additional parameter m. The authors give a heuristic for what a good choice of m might be; however if m is not chosen well, then the algorithm is liable to give an overly inaccurate model and take very long to compute, or fail.

The algorithm run time is O(d^2 n + dm log(m))---where generally m << n.

The algorithm is also justified by experiments.


**Summary Of The Review:**

The paper's goal is to design a user friendly differentially private linear regression algorithm. The solution provided seems to strike a good balance between user-friendliness and accuracy according to theoretical results and experiments. I don't think this algorithm will be the definitive word on user-friendly DP LR, but it is a good step forward in an area that addresses a question at the intersection of theory and practice.

NOTE: I have not verified the correctness of all the claims and I did not read the appendix. In "Correctness" below, I am indicating that I believe the theorems and they seem to be well justified in the appendix (which I did not read).

---

> ### Author Response · Authors · 2022-11-10
> **Response to reviewer Ndnv**
>
> Thanks for the detailed review. We've highlighted questions/comments and attempted to answer them below. Please let us know if you have further questions.
>
> > The possibility that the algorithm fails to return a model for some reasonable specifications of m seems annoying.
>
> Unfortunately, failure is an intrinsic cost of propose-test-release algorithms like the one suggested here. We suggest that the high success rate of our algorithm for datasets satisfying the sketched heuristic partially mitigates this. Another possible mitigation is to take the half of the privacy budget left over after a failed PTR check and pivot to running one of the baseline algorithms (AdaSSP or DPSGD). Since this is a less clear algorithm conceptually, we did not discuss it in the paper, but it might be a reasonable option in practice.
>
> > The time complexity suggests that the method is not well suited for high-dimensional data.
>
> The time complexity is $O(d^2n + dm\log(m))$. Note that $O(d^2n)$ is the cost of OLS without privacy, and for the reasonable range of $1 \leq m \leq n/d$, the $dm\log(m) < d^2n$ unless $n \gg 2^{d^2}$.  Moreover, given some other regression method taking time $S(n, d)$, as long as its dependence on $n$ is linear, using the method in our algorithm will result in overall time $O(S(n, d) + dm\log(m))$. As a result, we suggest that our algorithm does not add meaningful time complexity over generic OLS, or other regression methods, except in very extreme parameter ranges.

---

> > ### Comment · Reviewer_Ndnv · 2022-11-18
> > **Response**
> >
> > Thanks for addressing the time complexity concern.
> >
> > Regarding the possibility of failure: I understand that possible failure is intrinsic to propose-test-release algorithms. However, the problem could be solved by different types of algorithms---perhaps some do not have this intrinsic deficiency. It is nice to see the empirical work on some problems that suggests the algorithm can work effectively in practice.

---

### Official Review · Reviewer_Nbxy · 2022-10-25

**Confidence:** 3
**Clarity, Quality, Novelty And Reproducibility:** 1. This paper is written well.
2. The…
**Correctness:** 4
**Technical Novelty And Significance:** 3
**Empirical Novelty And Significance:** 4
**Recommendation:** 8

**Strength And Weaknesses:**

Strengths: 1. This paper provides the first practical algorithm for DP linear regression based on high dimensional median. The emprical experiments justified the effectiveness of the proposed algorithm
2. It is kind of surprising to see that this type of algorithm works well on real data. For example, the theoretical results in Brown at el 2021 assume that the data is exactly Gaussian and not able to generalize to sub-gaussian. This paper provides strong evidence and shows that this type of approach could be used for real data.
Weaknesses: 1. Unlike prior works, no utility guarantees are provided here for Gaussian data using approximate tukey depth.
2. A better DP-SGD baseline would be (Varshney at el 2022). The difference is that the clipping threshold is adaptively estimated from the data, which is also easy to implement. Theoretically, (Varshney at el 2022) gives near-optimal rates for sub-gaussian-like data.



Questions:
1. I guess the proposed approach could be first used for mean estimation directly.  Is this true? If this is true, how is the empirical performance?
2. To my understanding, this approximate tukey depth is similar as coordinate-wise median. Is there any hope to provide theoretical analysis for gaussian data under $\ell_\infty$-norm?
3. I guess the proposed method could also provide some robustness against the corruption of the datasets as a side product of using high dimensional median (as in Liu at el 2021 or Liu at el 2022). Can this be verified through experiments or theoretical justifications?

**Summary Of The Paper:**

This paper provides a practical algorithm for differentially private linear regression. Algorithmically, this proposed method is a combination of Theil-Sen estimator for linear regression and private mean estimator using Tukey median appeared in (Brown at el 2021) and (Liu at el 2021). Prior works have provable guarantees for gaussian data but are exponential time. The proposed method computes the median of OLS solutions using approximate Tukey depth and exponential mechanism. This paper does not have theoretical utility guarantees for gaussian data. But this paper provides extensive experiments to demonstrate the effectiveness.

**Summary Of The Review:**

This paper provides a practical algorithm for DP linear regression, which can be seen as the most fundamental task for learning with differential privacy. Unlike prior works in this domain, this paper also provides extensive experiments on real data.

---

> ### Author Response · Authors · 2022-11-10
> **Response to reviewer Nbxy**
>
> Thanks for the detailed review. We've highlighted questions/comments and attempted to answer them below. Please let us know if you have further questions.
>
> > I guess the proposed approach could be first used for mean estimation directly. Is this true? If this is true, how is the empirical performance?
>
> The PTR + restricted exponential mechanism approach can indeed be used for mean estimation directly, and was originally proposed for it by Brown+ 21. The main contribution of our work is making it computationally tractable and applying it to regression. We agree that our approach could also be applied to mean estimation directly, but we suggest that including experimental results for this separate problem is beyond the scope of this paper.
>
> > Unlike prior works, no utility guarantees are provided here for Gaussian data using approximate tukey depth.
>
> We speak to some of the challenges involved in getting such a guarantee in our response to reviewer 3oJ7.
>
> >  Is there any hope to provide theoretical analysis for gaussian data under $\ell_\infty$-norm?
>
> Is the suggestion that approximate Tukey depth can be viewed as exact Tukey depth under a different norm? If we understand correctly, equipping the space with a different norm does not (immediately) change the notion of half-spaces or cardinality of intersections with those half-spaces. However, we do not claim that an analysis in this direction is impossible.
>
> > A better DP-SGD baseline would be (Varshney at el 2022). The difference is that the clipping threshold is adaptively estimated from the data, which is also easy to implement. Theoretically, (Varshney at el 2022) gives near-optimal rates for sub-gaussian-like data.
>
> One drawback of the guarantees provided by Varshney+ 22 is that they assume significant a priori knowledge of the data distribution. For example, their utility guarantee for the adaptive clipping algorithm (Theorem 10) assumes that the end user can provide accurate bounds on several functions of the data and noise distributions (condition number, moments, etc.) and relies on these quantities to set several parameters in its algorithm. We suggest that this weakens the practicality of this algorithm relative to DPSGD. We also note that the performance of tuned DPSGD already presented in the submission should upper bound the performance of the adaptive clipping algorithm, since tuned DPSGD effectively tunes this clipping parameter without paying any privacy cost.
>
> > I guess the proposed method could also provide some robustness against the corruption of the datasets as a side product of using high dimensional median (as in Liu at el 2021 or Liu at el 2022). Can this be verified through experiments or theoretical justifications?
>
> Robustness likely depends on the nature of the data corruption. Lightweight experiments suggest that our method copes slightly better than the other DP baselines when the data includes a very small fraction of highly corrupted points (e.g., synthetic data where 0.1% of points have labels inflated by a factor of 100) and slightly worse when the data includes a larger fraction of moderately corrupted points (e.g., synthetic data where 10% of points have labels inflated by a factor of 5). A possible explanation is that in the first case the vast majority of models include no corrupted data, and the remaining models are essentially ignored; in the second case, most models include a corrupted point, and this has an outsize influence on models which are trained on a small number of points. Further investigating the robustness of our method may be a reasonable avenue for future work.

---

### Official Review · Reviewer_fjWP · 2022-11-01

**Confidence:** 2
**Correctness:** 3
**Technical Novelty And Significance:** 2
**Empirical Novelty And Significance:** 2
**Recommendation:** 5

**Clarity, Quality, Novelty And Reproducibility:**

The quality of the writing is good with some (not so surprising) novel aspects. Also, I am a little worried about no comparison with the prior work with respect to the utility guarantee. The paper seems to have a more theoretical feel to it, hence I would like to see a comparison on that front.

**Strength And Weaknesses:**

The algorithm is more efficient, does not require hyperparameters and does not requirement feedback from the user on the label norm or feature norms.

The paper seems to natural generalization of known results. In particular, they achieve the polynomial running time by approximating Tukey depth, whose exact computation is NP-hard. Moreover, the paper does not compare its utility guarantee with the known previous works. As such, it is not clear to me what is the advantage of this algorithm? Is it just in the terms of running time, but what about other parameters, like utility? Isn't practical applications would be more concerned with the utility of the algorithm over a run time improvement?

The paper is overall well written-- I did not get to check the proofs, but I would love to see them over the next month. I really would like to understand what is the technical challenges faced in this paper? Like, how does considering approximate Tukey depth changes the utility guarantee. What is the sensitivity of the approximate Tukey depth? I can understand that Tukey depth is 1-sensitive function, but why is the approximate version the same? I believe the approximation in multiplicative. In general, some care has to be taken when considering approximate functions (see the recent paper https://arxiv.org/abs/2210.03831 for some discussion on this front and their use of smooth sensitivity).

The proper credit of DP-SGD should go to this paper https://cseweb.ucsd.edu/~kamalika/pubs/scs13.pdf or BST paper. Abadi et al. just did the moment accountancy which was new. DP-SGD was known before. Likewise, Sheffet's paper gave the first analysis of statistical inference for OLS using differentially private estimators; however, it was not the first to study OLS.

Page 3. The last sentence has two full stops.

I am not sure how the authors can claim that the empirical distribution of models has fast tail decay! I would argue it is more heavy-tailed.

The first sentence of the second paragraph in Sec 3.2 does not make any sense.

I am surprised that the authors have used dataset that has very very small dimension when their central claim is that the algorithm is practical. No one is going a linear regression model for d=25.

**Summary Of The Paper:**

The current paper under submission studies linear regression and gives an algorithm that is "more practical" than the previous approaches. It is based on the approach of Alabi et al. to a large degree and uses known techniques in DP literature to achieve its objective.

**Summary Of The Review:**

Please see above.

I have not read the proof so I am not judging whether the claims of the papers are right or wrong. That part of my review is subject to change.

---

> ### Author Response · Authors · 2022-11-10
> **Response to reviewer fjWP**
>
> Thanks for the detailed review. We've highlighted questions/comments and attempted to answer them below. Please let us know if you have further questions.
>
> > … the paper does not compare its utility guarantee with the known previous works. As such, it is not clear to me what is the advantage of this algorithm? Is it just in the terms of running time, but what about other parameters, like utility? Isn't practical applications would be more concerned with the utility of the algorithm over a run time improvement?
>
> The main contribution of this paper is to demonstrate the empirical advantage, in terms of utility, of our algorithm over previous algorithms. While formal utility guarantees are important, we believe that the empirical success of the algorithm is a clear advantage over existing methods, including previous state-of-the-art algorithms such as AdaSSP. We also believe that the strength of the empirical performance should invite further analysis, including improved formal guarantees.
>
> We also reiterate that the primary advantage of this algorithm over existing work is its ability to obtain good empirical performance without 1) assuming that end users can specify accurate bounds on the data or 2) assuming that users can conduct extensive hyperparameter tuning that will not be counted toward the privacy budget. In contrast, prior works (AdaSSP and DPSGD, respectively) rely on these assumptions. This reduces the practicality of these prior algorithms, as users often struggle to provide a priori data bounds (Sarathy+ 2022, see bibliography) and hyperparameter tuning realistically costs extensive computational resources, time, and privacy budget. We note further that our algorithm's empirical utility typically matches or exceeds that of AdaSSP and DPSGD, even though they are given non-private access to additional information.
>
> > I really would like to understand what is the technical challenges faced in this paper? Like, how does considering approximate Tukey depth changes the utility guarantee. What is the sensitivity of the approximate Tukey depth? I can understand that Tukey depth is 1-sensitive function, but why is the approximate version the same? I believe the approximation in multiplicative. In general, some care has to be taken when considering approximate functions (see the recent paper https://arxiv.org/abs/2210.03831 for some discussion on this front and their use of smooth sensitivity).
>
> The main technical challenge in this paper is constructing a DP algorithm for linear regression that, unlike prior work, does not rely on strong a priori knowledge of the data or carefully (and non-privately) chosen hyperparameters to obtain reasonable utility. Doing so required borrowing and adapting tools from recent work on DP mean estimation (Brown+ 22) to be computationally tractable. Obtaining a tight approximation to Tukey depth is not a goal of the paper. We instead use approximate Tukey depth because it is a computationally tractable notion of depth that still captures the intuitive notion of selecting a "central" point while remaining 1-sensitive. The proof of 1-sensitivity is short: the score is a minimum over d 1-dimensional depths, and 1-dimensional depth is 1-sensitive. Note that the privacy of the mechanism does not depend on how well approximate Tukey depth approximates exact Tukey depth.
>
> > I am not sure how the authors can claim that the empirical distribution of models has fast tail decay! I would argue it is more heavy-tailed.
>
> Figures 7-14 in the appendix plot the empirical model distributions (blue bars) and overlay Gaussians with the same mean and variance as the empirical distribution (red curve). Our claim essentially reduces to the observation that the empirical distribution's tails lie under those of the overlaid Gaussian.
>
> > The first sentence of the second paragraph in Sec 3.2 does not make any sense.
>
> The first sentence of the second paragraph in Section 3.2 is "The overall strategy applies work done by Brown et al. (2021)." This sentence appears correct to us. Can you elaborate on what looks confusing about it?
>
> > I am surprised that the authors have used dataset that has very very small dimension when their central claim is that the algorithm is practical. No one is going a linear regression model for d=25.
>
> We agree that extending these results to a larger range of d would improve the applicability of our method. However, the baseline results for d < 25 demonstrate that even this relatively simple setting has not been solved by existing methods without making strong assumptions on end user knowledge of the data or end user private hyperparameter tuning capabilities. We suggest that the results obtained by our method for d < 25 without these strong assumptions constitute a promising starting point, and that extending these results to larger d is a good next step for future work.

---

> > ### Comment · Reviewer_fjWP · 2022-11-22
> > **Response to rebuttal**
> >
> > I apologize for the delay. I will try to be brief.
> >
> > I do not understand how the authors can say that the main objective of the paper is to get "empirical advantage" (or that the focus was not on developing good theory) and still only care about d=25 or less. If the authors' work is empirically advantageous, then I would suggest running it for higher dimension. Scalability (and the curse of dimensionality) impedes the practicality of many solutions. So, I do not feel it is fair to compare a theoretical result with their experimental focus paper and then not give convincing experiments.  What to guarantee me that the algorithm behaves well, is stable, and produces good results, only for a small value of d, and is completely unstable or non-tractable as d increases!
> >
> > I fail to understand the grammatical meaning of the sentence: "The overall strategy applies work done by Brown et al. (2021)."
> >
> > For heavy-tailed vs light-tailed, I believe a lot of works have recently shown that heavy-tailed phenomenon is more common. I am fine with making an assumption of the concentrated tail, but an assumption is only good if it is backed up in practice, especially when critical concepts as privacy is at play. I would not bet on a scheme that is based on subset-sum, but I would bet on a scheme based on factoring assumption even though subset-sum is hard in the worst case.

---

> > > ### Author Response · Authors · 2022-11-23
> > > **Re: Response to rebuttal**
> > >
> > > Thanks for the response! We'll try to address each point in turn.
> > >
> > > Strength of the Contribution: We appear to disagree over the relevance of private regression algorithms for moderate-dimensional data (d < 25). We suggest that linear regression in this setting is such a basic statistical primitive that the absence of existing DP solutions for it that do not rely on "cheating" (e.g., assuming strong bounds, looking at the data, and hyperparameter tuning) is a nontrivial gap in the literature and a significant weakness for usable DP. The current work fills this gap. We agree that extending the current work to also cover high-dimensional settings would paint a more complete picture, but we disagree that the moderate-dimensional case is not worthy of study in its own right.
> > >
> > > Grammar: "The overall strategy applies work done by Brown et al. (2021)" means that we used methods that were developed by Brown et al. within this work.
> > >
> > > Tails: It may help the discussion to clarify the meaning of "heavy-tailed". A heavy-tailed data distribution is indeed common, but this is distinct from a heavy-tailed model distribution; the former concerns the spread of the data points themselves, while the latter concerns the spread of models trained on subsamples of the data. In particular, a heavy-tailed data distribution does not imply a heavy-tailed model distribution. Figures 7-14 in the paper demonstrate that the model distributions arising from the paper's datasets are, empirically, not heavy-tailed. Finally, we emphasize that our method's differential privacy guarantee in no way relies on any assumption about the data. We only rely on the distributional assumption to make formal guarantees about the method's accuracy; privacy always holds.

---

### Decision · Program_Chairs · 2023-01-20

**Decision:**

Accept: poster

**Justification For Why Not Higher Score:**

There are certain disadvantages to this work, including a heuristic selection of some parameters and the fact that the experimental results are in fairly low dimension.

**Justification For Why Not Lower Score:**

Based on the reviews, scores, and my own reading, the paper appears to be an interesting contribution.

**Metareview: Summary, Strengths And Weaknesses:**

This paper gives a method for differentially private linear regression that is simpler and claimed to be more practical compared to previous approaches. A key idea is leveraging the Tukey median -- an intractable to compute multivariate generalization of the median with good robustness properties. The authors propose some heuristic approximations to the Tukey median that seem to give good practical performance, and use standard tools to privatize it. Overall, the reviewers considered this submission to be a solid contribution that is appropriate for ICLR.

**Note From Pc:**

if the above contains the word "oral" or "spotlight" please see: "oral" presentation means -> notable-top-5% and "spotlight" means -> notable-top-25%. As stated in our emails, we are disassociating presentation type from AC recommendations